# Can Water-Detection Indices Be Reliable Proxies for Water Discharges in Mid-Sized Braided Rivers Using Coarse-Resolution Landsat Archives?

**Peng Gao** [1,2,*] [iD], **Barbara Belletti** [3] [iD], **Hervé Piégay** [3] [iD], **Yuchi You** [4] and **Zhiwei Li** [5] [iD]

1    Department of Geography and the Environment, Syracuse University, Syracuse, NY 13244, USA
2    EUR H$_2$O'Lyon, University of Lyon, Site of ENS, 15 Parvis R. Descartes, F-69362 Lyon, France
3    UMR 5600 CNRS—EVS, University of Lyon, Site of ENS, 15 Parvis R. Descartes, F-69362 Lyon, France;
     barbara.belletti@cnrs.fr (B.B.); herve.piegay@ens-lyon.fr (H.P.)
4    College of Water Resources & Civil Engineering, Hunan Agricultural University, Changsha 410128, China;
     youmandy@163.com
5    State Key Laboratory of Water Resources and Hydropower Engineering Science, Wuhan University,
     Wuhan 430072, China; lizw2003@whu.edu.cn
*    Correspondence: pegao@syr.edu

**Abstract:** The use of water detection (WD) indices to infer daily discharge ($Q_d$) has a great potential to enrich needed hydrological data for understanding fluvial processes driving the morphological changes of braided rivers. However, no consensus has been reached on which one stands out for use in mid-sized braided rivers. In this study, we compared the physical characteristics of three most commonly used WD indices, the Normalized Difference Water Index (NDWI), Modified Normalized Difference Water Index (MNDWI), and Normalized Difference Moisture Index (NDMI), for two mid-sized braided reach segments in the Qinghai-Tibet Plateau, China, that have different morphological structures. Relying on the Google Earth Engine web interface, we calculated the total mean water width (*WWt*) based on the detected surface-water areas ($A_s$) and braiding index (*BI*), as well as the mean values (m) of these indices over about four decades at the braided corridor scale (cs) (mNDWIcs, mMNDWIcs, and mNDMIcs). We then examined different responses of these indices to water and non-water features and their best threshold values for characterizing channel structures. Our analyses demonstrated that (1) NDWI and MNDWI perform well for detecting braided channel structures with the threshold of zero; (2) *WWt* is generally better correlated to $Q_d$ in a linear style than WD indices do, particularly when calculated from MNDWI; and (3) among WD indices calculated at the braided corridor scale, mMNDWIcs shows a better relationship with $Q_d$ than mNDMIcs does. Finally, we provided mechanisms that may explain these differences in terms of photometric discrepancies in calculating *WWt* and *WD* indices and the impact of image resolution on their calculations.

**Keywords:** water-detection index; surface-water area; total mean water width; braiding index; daily discharge





## 1. Introduction

Braided rivers have the most dynamic planforms, compared to those of other river types (i.e., straight, meandering, and anabranching rivers), mainly because of their high sediment supply and low resistance [1]. The ability of capturing morphodynamic processes of braided rivers is contingent on the availability of water discharge values, which allow for comparing them with braiding intensity through time and understanding patterns of the temporal changes in braided flowing channels in response to sediment and discharge inputs [2]. Unfortunately, braided rivers located in remote mountain regions often have no or very limited daily discharge ($Q_d$) data [3,4]. The lack of $Q_d$ data has become a fundamental obstacle in fully understanding fluvial mechanisms driving changes in braided river morphology and developing better water management practices.

Remotely sensed data have long been used to infer water discharges in various types of rivers, including braided rivers [5–9]. The inference is rooted in surface water detection based on the photometric principle that each spectral band of optical remote sensing imagery has different characteristic reflectance intensities for different types of land covers [10]. The key is to separate image pixels representing water from others. Among a wide variety of methods developed so far, a commonly used suite is distinguishing water elements of images by classifying water-detection indices, including, for example, the Normalized Difference Vegetation Index (NDVI), Normalized Difference Water Index (NDWI), Modified Normalized Difference Water Index (MNDWI), and Normalized Difference Moisture Index (NDMI) [11–15]. These indices are developed from the combination of two individual spectral bands (i.e., green, red, near-infrared (NIR), or short-wave infrared (Swir)) and have standardized values between −1 and 1. Although other indices, for instance, the automated Water Extraction Index [16,17], have also been created, they are less frequently used possibly because their values are not standardized, obstructing their general application.

Relying on different threshold-based classification schemes, the developed water-detection indices enable extracting different water features from satellite images, such as surface water extents [18,19]. These features can be subsequently used to estimate water discharges in rivers with shallow bars (e.g., braided rivers) [20,21]. The approaches of determining water discharges may generally fall in the theoretically based and the empirically based schools. In the first school, water discharges may be calculated by extracting mean river widths, mean flow depths (or cross-section areas), and mean flow velocities from remotely sensed images [22–24]. They may also be predicted over a consecutive period by (i) combining remotely sensed data with the Mass-conserved Flow Law Inversion (McFLI) approach [19,25] or (ii) incorporating satellite data into hydraulic models or at-many-station hydraulic geometry as key input [26–28]. The second school is essentially about developing different types of rating curves, in which water discharges are correlated to and empirically determined by different water features, such as flow widths, water stages, or surface-water areas [29–31]. Among these, a common approach resorts to the flow width, which may be easily extracted from satellite image archives. Rating curves, in which water discharges ($Q$) are correlated to the associated flow widths ($W$), are empirically determined, following the classic equation $Q = aW^b$. While these methods have greatly increased the availability of river discharges globally, they are often developed for large rivers (channel width > 100 m) where the flow width varies significantly with discharges and these variations can be easily measured from remote sensing images [32,33]. However, the flow-width approach may not be appropriate for small and medium-sized rivers [34] because channel width cannot be extracted from the primarily available archives of Landsat images with 30 m or coarser spatial resolutions.

A general geomorphological feature of the braided rivers is that they flow within a fairly wide gravel corridor, within which multiple flow channels have large width/depth ratios [1,35,36]. Therefore, the flow width is most sensitive to the change in flow rates [37,38] and is the best water feature to link to the associated water discharge [39]. However, extracting flow widths from satellite images based on water-detection indices is only possible if channel widths are large enough compared to the image resolution. Unfortunately, this is not the case for many braided channels, even in many larger braided corridors.

A recent study in a braided river reach on the northeastern side of the Qinghai-Tibet Plateau [4] suggested that some water-detection indices (e.g., mean index values within a braided active channel area) can be correlated to the water discharges of braided rivers. This study shows promise of using remote sensing data with a coarse resolution to capture the hydrological variability of mid-sized braided rivers. The braided active channel is defined here as a corridor of unvegetated bars and all flowing channels [40].

An additional but important advantage of directly resorting to water-detection indices for discharge estimation lies in the fact that emerging techniques in cloud-based computer platforms, such as Google Earth Engine (GEE) [41–43], have made it possible to automate the process of extracting these indices as almost continuous through time series from a large

number of available satellite images over decades and hence greatly improve the efficiency of determining water discharges using these indices. It can be envisioned that in braided rivers where obtaining water discharges is challenging owing to physical constraints and financial limits, establishing rating curves between braided corridor water-detection indices and water discharges has a great potential of successfully and rapidly obtaining continuous water discharges for better understanding morphodynamics of these braided rivers.

In this context, our study aims to (1) test the behavior of different mean water-detection indices calculated within braided corridors compared to mean water width based on the classic thresholding of water surface area across a range of water discharges, (2) identify the water-detection index that is best linked to water discharges, and (3) interpret the potential differences between these indices.

## 2. Materials and Methods

### 2.1. The Studied Braided Reaches

Braided rivers constitute a major type of river patterns on the Qinghai-Tibet Plateau (QTP). They are widely distributed in the alluvial valleys whose elevations vary between 3400 and 4600 m. We selected two braided rivers in the QTP because they are quite large compared to other regional settings and also with greatly varied braided structures in terms of flow channels and flow architecture. The two specific braided reach segments are located within the Lhasa River and the Upper Lancang River watersheds, both of which are equipped with a gauging station (Figure 1a). The first one (Lh_Seg) extends along the middle reach of the Lhasa River (29°42′50.25″N, 91°25′13.02″E). This segment is dominated by a semi-arid climate featuring a short, warm, and wet season from June to September and a long, cold, and dry season from October to May and has been subject to intense human disturbance including agriculture development, mining activities, urban sprawl, and dam construction [44]. The Lh_Seg covers a river section about 10 km long (Figure 1b) and is located about 20 km upstream of Lhasa City, the capital of the Tibet Autonomic Region, China. A gauging station about 35 km downstream of the Lh_Seg (Figure 1b) has produced limited daily discharge data in the 1987–2002 and 2008–2016 periods that can reasonably well represent hydrological variations within the selected segment. The braided channels within the Lh_Seg are developed in a valley extending laterally about 3.46 km and are separated by gravel and sandy bars dotted with artificially planted trees (Figure 1d). The braided corridor is about 1.6 km wide, and the wetted channels at low flow can range from a few meters to 100 m in width.

The second segment (32°08′49.67″N, 96°32′25.68″E) (Lc_Seg) is located in the middle reach of the Upper Lancang River (Figure 1c). The Lc_Seg spans about 6 km and is located downstream of a local town (Nangqian) (Figure 1c). It belongs to a region that is dominated by a typical monsoon climate with a distinct wet season between June and October and a dry season from November to May [4]. The braided branches are mixed with sandy gravel bars, many of which survive relatively dense shrubs and small trees (Figure 1e). The braided corridor is less than 1 km wide, and the wetted channels at low flows are less in number but wider compared to those in the Lh_Seg. The nearest gauging station is about 10.6 km upstream of the Lc_Seg. Thus, its available daily discharges spanning over the 1989–2017 period were used to represent the hydrological variations of the selected braided segment.

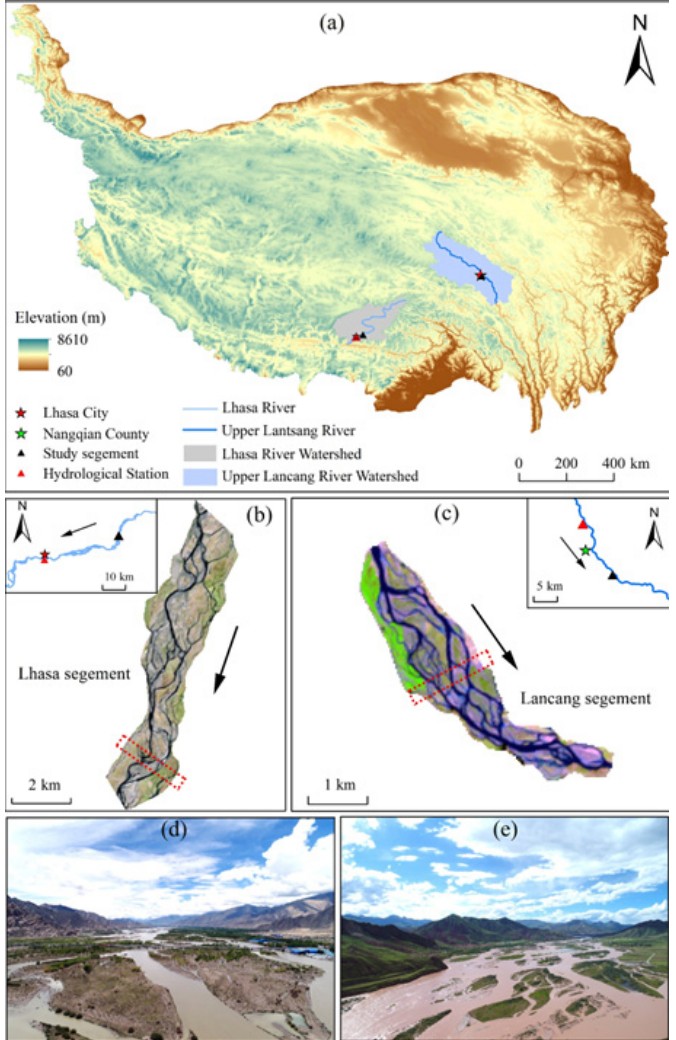

**Figure 1.** Geographic settings of the study area and the selected braided reach segments. (**a**) Locations of the two selected braided segments in the Qinghai-Tibet Plateau and their watersheds (**b**) and (**c**) the planform view of the two braided segments (e.g., the Lh_Seg and Lc_Seg sites) and their associated hydrological stations; (**d**,**e**) an oblique view of the two selected braided corridors (for locations, see the red rectangles in (**b**,**c**)).

*2.2. Methods*

Three commonly used water-detection indices (i.e., NDWI, MNDWI, and NDMI) were selected in this study. NDWI is based on the green and near-infrared (NIR) bands [11], MNDWI uses the green and short-wave infrared (SWIR) bands [15], while NDMI relies on the NIR and SWIR bands [14]. Although NDVI, which is based on the NIR and red bands, is often used to detect vegetation coverage [45], it is extracted here to help understand the nature of the three selected indices. In general, image pixels with positive values of these indices represent water area and/or extent [46,47], though a negative threshold value of some indices has also been chosen for detecting water features [47,48].

Google Earth Engine (GEE) was used to extract the values of the original bands (i.e., green, red, NIR, and SWIR) for determining the four indices from the entire archive of Landsat image collections between the 1980s and 2021 (https://developers.google.com/earth-engine/datasets/catalog/landsat, accessed on 29 November 2023), including Landsat 4 (1982–1993), 5 (1984–2012), 7 (1999–2021), and 8 (1999–2021). These extracted values represent the atmospherically corrected surface reflectance at level 2 (i.e., Collection 2, Tier 1). The GEE data processing involves four steps: first, identifying all bands from

all collections and assembling them in a single collection to ensure that the images cover at least 98% of the studied sites and have a cloud cover value lower than 20% (over the entire images); second, calculating the normalized difference water and vegetation indices and adding them to the image collection as new bands; third, directly extracting statistics (i.e., mean, median, and standard deviation) of the four indices within each of the selected braided corridors; and fourth, classifying water features based on the threshold of zero for NDWI, MNDWI, and NDMI, from which water surface areas and water perimeters were calculated. These values were subsequently exported in the .csv format for further analyses. Notably, the mean total mean water width (*WWt*) and the braiding index (*BI*) were further calculated. The *WWt* is the effective width obtained by dividing the water surface area by the braided corridor length. It represents the inundation area averaged over the whole reach spread in one to several individual channels, rather than the width observed in a single-bed channel (see also [5]). The *BI* is defined here as the total sinuosity index [49], obtained by dividing the half perimeter of the surface waterbody by the reach length. The calculation error for these metrics depends on the threshold method adopted and on the choice of the threshold value (here water-detection index > 0), as well as on the resolution of the source images (here 30 m), but can be considered negligible for the purposes of the work.

Using the extracted values of total water width from the thresholding of the surface waterbody and the three mean indices calculated within each selected braided corridor segment, which were denoted as mNDWIcs, mMNDWIcs, and mNDMIcs, and the available daily discharge ($Q_d$) corresponding to the acquired images, we performed a set of analyses to reveal the nature of each index in identifying water features.

We first illustrated the distributed values of each index in the entire Lh_Seg and visually assessed the sensitivity of different water-detection thresholds for four selected $Q_d$ values. These four $Q_d$ values reflect the range of hydrological variations in the Lh_Seg, which were 19.6, 129, 255, and 948 m$^3$/s. The dates of the images associated with these discharges are 2/23/2016, 11/26/2000, 6/26/1988, and 8/30/2009, respectively. The highest $Q_d$ represents the annual discharge in the Lhasa River. Higher discharges were not selected mainly because there were no appropriate images available to match them. Then, we investigated the detailed responses of index values in the Lh_Seg to water and non-water areas under these four different hydrological conditions. This investigation began with selecting three transects representing the upstream, middle, and downstream parts of the braided segment in each image (Figure 2) and generating variation patterns of these index values over the transects of all four images. Along each transect, pixels were classified, based on photo interpretation, into water (i.e., wet channels) and non-water, which mainly includes dry channels and lands (e.g., bars with and without vegetation, braidplains, and roads). Patterns of variation for each index along these transects in each of the four images were subsequently examined to identify the performance of these indices regarding water bodies and non-water features.

Finally, we examined the whole data set that is available in terms of discharge, braided index (*BI*), and total mean water width (*WWt*) obtained using the surface-water area ($A_s$) from the three selected indices for the two sites, Lh_Seg and Lc_Seg:

$$A_S = \sum_{i=1}^{n} (Pi_{\text{index}} > 0) \times Pi_{res} \tag{1}$$

$$WWt_{\text{index}} = A_S / L \tag{2}$$

$$BI = \sum_{i=1}^{n} (1/2\ P) / L \tag{3}$$

where $Pi_{\text{index}}$ corresponds to the pixel values of each assessed index, $Pi_{\text{res}}$ is the pixel resolution, which is 30 m in this case, the mean segment length $L = 6$ for Lc_Seg and 10 for Lh_Seg, and $P$ is the perimeter of the surface waterbody.

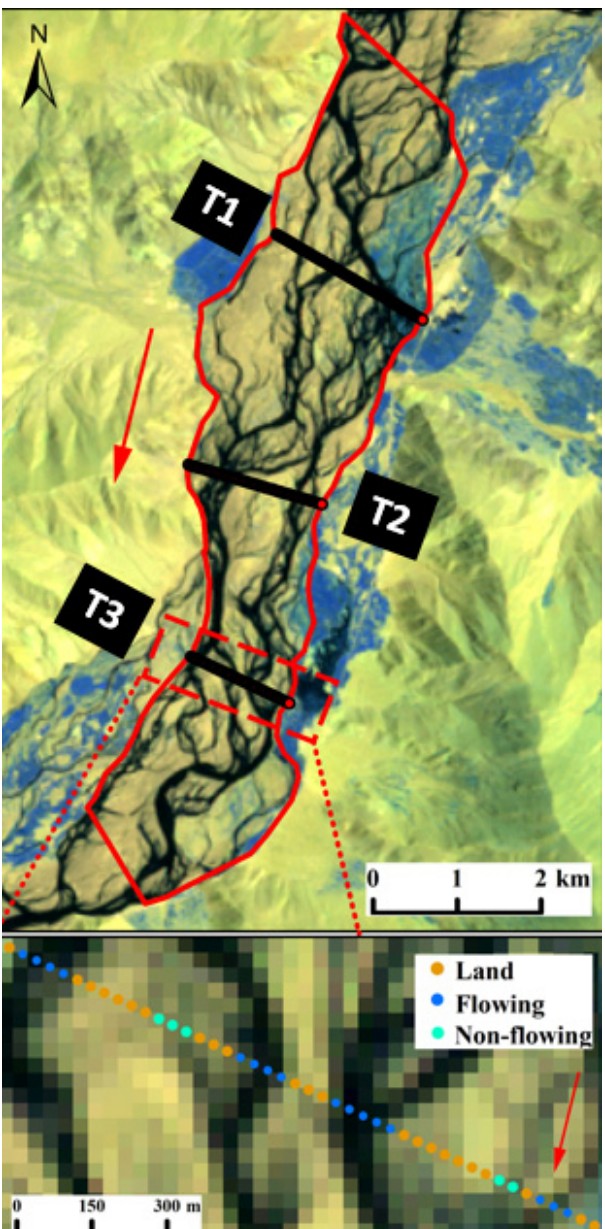

**Figure 2.** Positions of the three selected transects within the Lh_Seg site (T1–T3). The image was associated with the daily discharge of 948 m$^3$/s obtained on 06/26/1988. The red arrow refers to the flow direction. The red boundaries correspond to the analyzed braidplain extent (i.e., the braided corridor). The zoomed-in section at the bottom shows the pixels along T3 that are classified as land, wetted channels (i.e., flowing), and dry channels (non-flowing).

We then analyzed the time series of the mean index values within the two braided corridors. The examination was aided by comparing their basic statistical characteristics, which are means and relative standard deviations (RSDs). The RSD is the same as the coefficient of variation by magnitude but always remains positive. Then, we compared the correlations between *WWt* or mean index values in braided corridors and the associated $Q_d$ for all three indices of the two sites. Finally, we performed parallel analyses (time series and $Q_d$ relationships) for the extracted values of the three selected indices.

## 3. Results

### 3.1. Characteristics of Water-Detection Indices

The false-color images of the Lh_Seg site based on the three indices (MNDWI, NDWI, and NDMI) were generated at the four different $Q_d$ levels (Figure 3). For MNDWI and NDWI, the braided channels are identifiable in images with a wide range of water discharges (the first and second images of each series at each discharge in Figure 3), allowing for identifying water features, regardless of hydrological conditions. Between these two indices, MNDWI apparently presented the braided channel structure clearer than NDWI did for lower discharges. For NDMI, only a few main braided branches were able to be identified in the images with lower discharges, whereas all branches cannot be recognized in the images with higher discharges (the third image of each series at each discharge in Figure 3). This suggests NDMI tends to respond to water bodies and non-water features similarly, likely indicating overall humid conditions within the braiding corridor during the wet season.

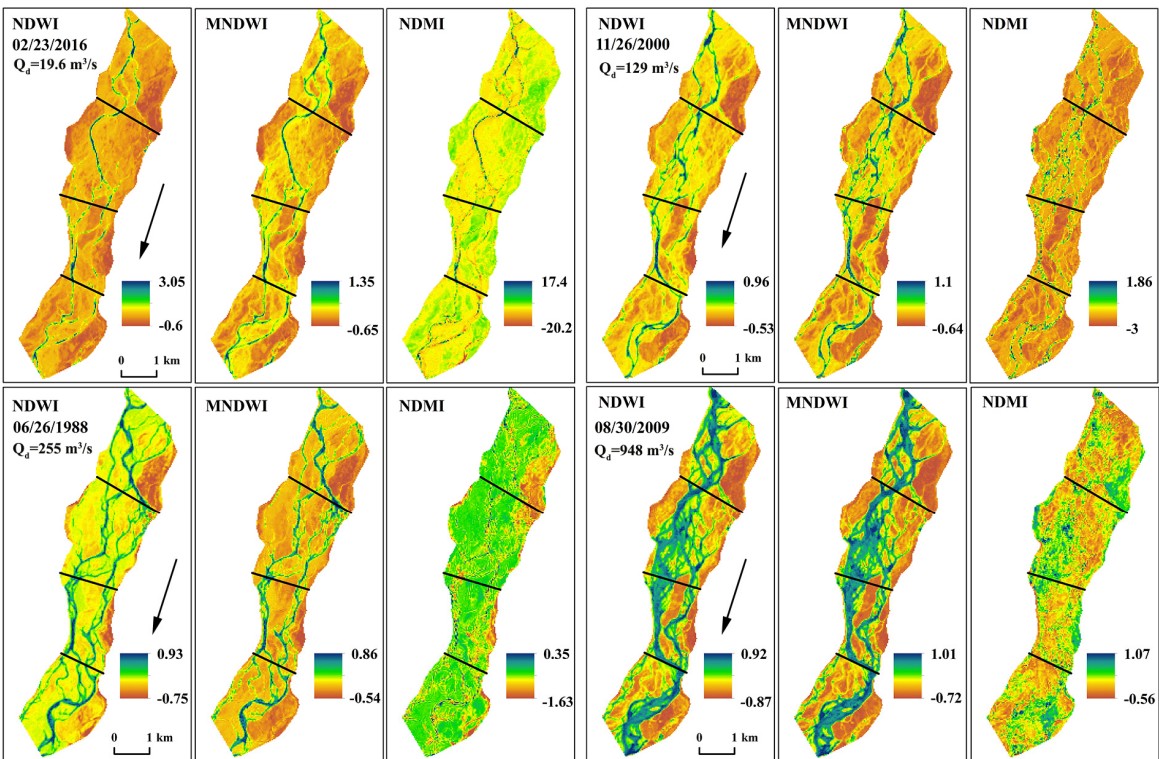

**Figure 3.** False-color images representing the values of each selected index over the entire area of the Lh_Seg site under the four different water discharges.

Using different thresholds for the three indices, the detected water bodies and braided network varied greatly regardless of the hydrological conditions (Figure 4). For any given discharge, the threshold chosen between −0.2 and 0.2 with the increment interval of 0.1 displays a different braided network (Figure 4). The threshold value of 0, which has been a common practice despite being criticized [10], produces consistent and realistic braided channel networks for different discharges, particularly for MNDWI and NDWI. Consistent with the observations shown in Figure 3, NDMI does not correctly represent the braided corridor at any of the analyzed thresholds, either underestimating (low discharge) or overestimating (higher discharge during the wet season) the wetted channels.

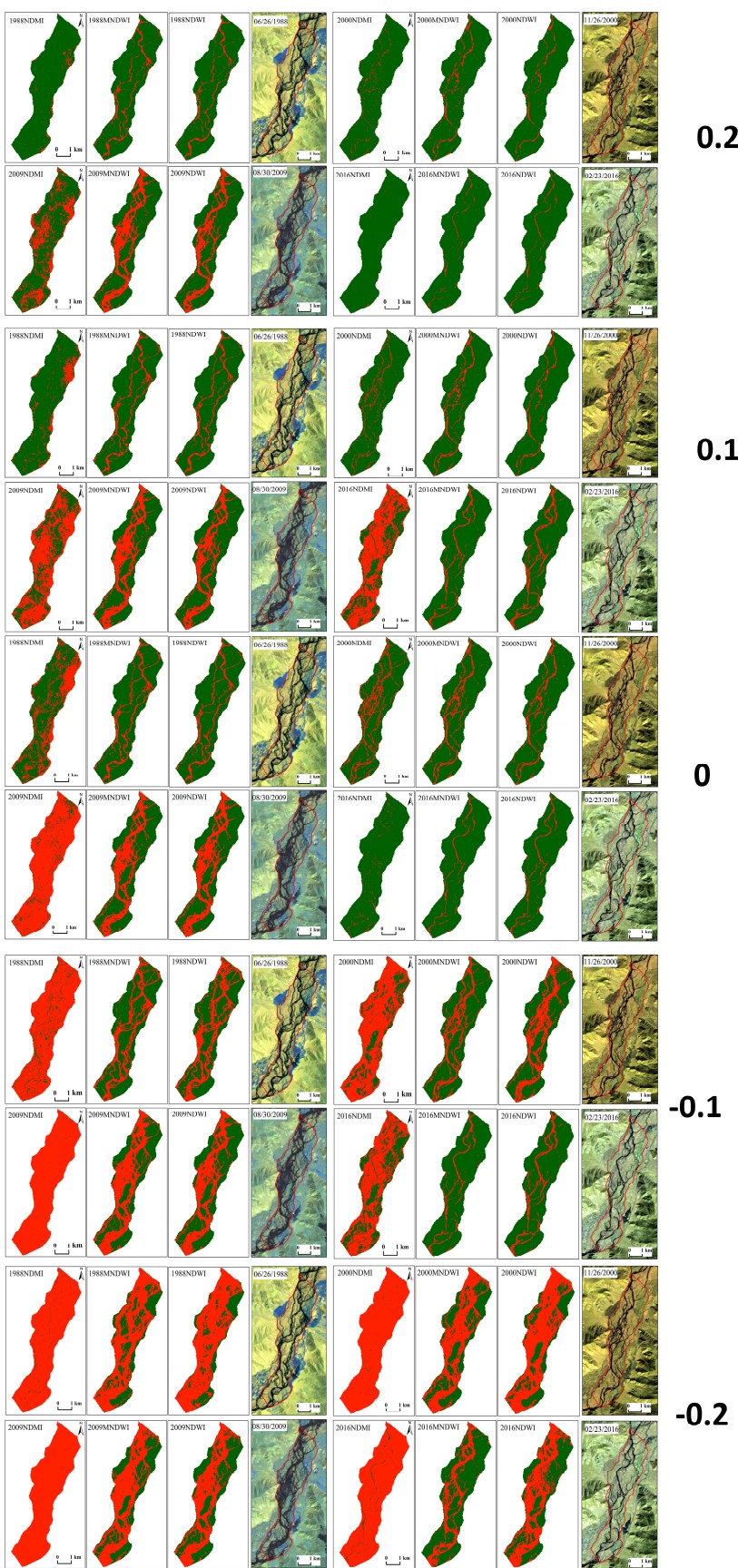

**Figure 4.** Surface-water areas (in red) represented by each index for the four different water discharges based on five different threshold values.

The detailed behavior of index values along the three transects and their represented planform sizes of the wetted channels (in m, obtained from the 30 m pixels) were demonstrated for the four indices under the four different water discharges along the Lh_Seg (Figure 5). In the top panes corresponding to a very low $Q_d$ (i.e., 19.6 m$^3$/s), NDWI (the purple curve in Figure 5) and MNDWI values (the green curve in Figure 5) were much higher in the wetted channel sections than those in the remaining parts (i.e., dry channels and lands) along all three transects (i.e., T1, T2, and T3). A wider channel branch tended to have higher index values than a narrower one. Variations in NDWI and MNDWI were small in the non-water sections and large in the wetted channels. This observation was confirmed by the two-sample difference test for water and non-water pixels along all three transects, which showed that their means were statistically different (*p*-value < 0.05).

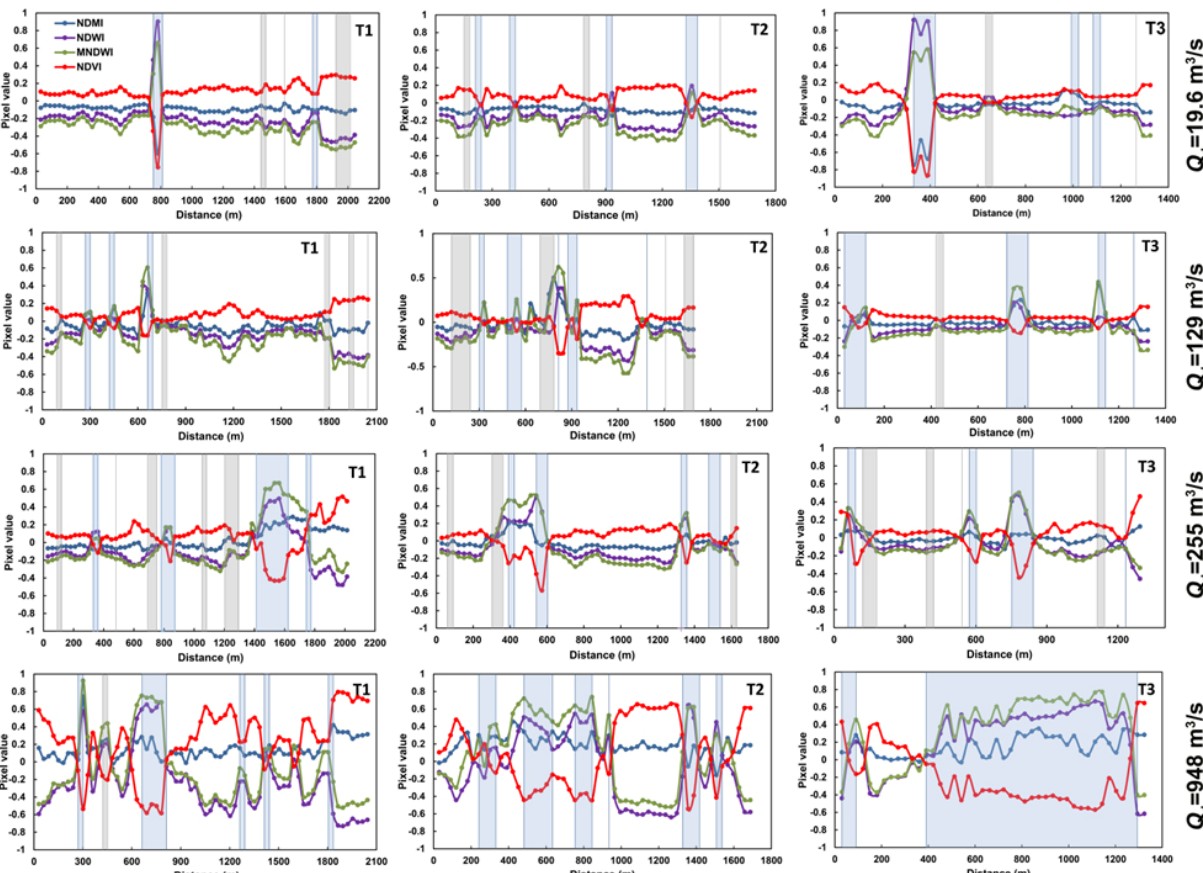

**Figure 5.** Changes in the values for the four indices along the three transects in the Lh_Seg site under four different water discharges. The curves begin from the right edge of the transect and end at the left edge (see Figure 2 for the orientation of each transect). The blue vertical bars represent wetted channels, the gray bars reflect the dry channels, and the remaining parts denote lands. Each row represents the results along the three transects under the given discharge.

NDMI (the blue curve) marginally decreased along T1 but dropped drastically in the first flowing channel. It remained similar between the wetted channel and non-water sections in T2 but became negative in the wider wetted channel along T3 (the top row in Figure 5). The mean values of NDMI for water and no-water pixels along all three Ts are weakly different at the 5% confidence level. Along the same three Ts, NDVI (the red curve) demonstrated lower or negative values in the wetted channels and higher values in the non-water sections (the top row in Figure 5). Its two types of means are different with statistical significance at the same confidence level. At this low hydrological stage, NDWI and MNDWI values for the water areas were clearly higher than those for the non-water

areas, whereas NDVI values had a reversed pattern. In contrast, NDMI values remained similar between the two types of surface features but displayed, locally, opposite behavior to that of NDWI and MNDWI in the wetted channels. The image having $Q_d$ of 129 m³/s had similar patterns of NDWI, MNDWI, and NDVI to those at the low $Q_d$ (the second row in Figure 5). NDMI, however, had higher values in the wetted channels than those in the non-water sections along all three Ts, which is opposite to its pattern at the lower discharge (i.e., $Q_d$ = 19.6 m³/s).

In the image associated with a medium-high $Q_d$ of 255 m³/s, NDWI and MNDWI retained similar patterns to those in the lower discharges, but the former tended to be lower in the wetted channels than the latter (the third row in Figure 5). This pattern was similar to that for $Q_d$ = 129 m³/s (the second row in Figure 5) but opposite when $Q_d$ = 19.6 m³/s (the first row in Figure 5). NDMI showed inconsistent patterns, higher in the wide wetted channel along T1 and one channel along T2 than in other sections but similar in other wetted channels to their neighboring values. At this hydrological stage, the mean values of all four indices between water and non-water pixels are different with strong statistical significance.

The image having the highest $Q_d$ (i.e., 948 m³/s) displayed similar general patterns of all indices along the three Ts to those for $Q_d$ = 129 m³/s but exhibited two distinct features (the bottom row in Figure 5). First, NDMI values in the wetted channels were mostly similar to those in the non-water sections, though their means are still statistically different at the 5% confidence level. In particular, their values in the very wide wetted channel along T3 were marginally greater than those on the lands. Second, NDVI values tended to be higher and always positive in the non-water sections along all three Ts, compared with those in the lower discharges, but were similar in the water pixels. Statistically, the means of NDVI, NDWI, and MNDWI between water and non-water pixels are all different with a strong statistical significance.

### 3.2. Temporal and Hydrological Characteristics of the Detected River Morphological Metrics

#### 3.2.1. Characteristics of $Q_d$, *BI*, and *WWt* in the Two Braided Segments

In the Lh_Seg site, 224 $Q_d$ values ranging from 19.6 to 948 m³/s were able to match the extracted Landsat images, and 75% of these values were less than 100 m³/s (Figure 6a). In the Lc_Seg site, 150 $Q_d$ values were obtained with the variation between 20.8 and 478 m³/s (Figure 6b). Although the maximum $Q_d$ in the latter was lower than that in the former, the $Q_d$ in both sites had similar frequency distributions (Figure 6a,b). The channel system in Lh_Seg has the mean braiding index (*BI*) of 3.6 based on MNDWI (Figure 6c), which is slightly higher and displays less variability than that in the Lc_Seg site (i.e., 2.5) (Figure 6d). The mean total water width (*WWt*) in the Lh_Seg site, which is 0.25 km (Figure 6e), is also slightly higher than that in the Lc_Seg site, which is 0.21 km (Figure 6f).

If considering the median conditions, we observe 1.9 and 2.6 m in channel width for 1 m³/s in Lc_Seg and Lh_Seg, respectively (Figure 6g,h). The channel widths of the Lh_Seg have experienced much more variability than those in the Lc_Seg under low-flow conditions. This might reflect the fact that the Lc_Seg has wider channel branches with less distributed flows than the Lh_Seg does, so it could be less sensitive to detection issues associated with the low resolution of Landsat images.

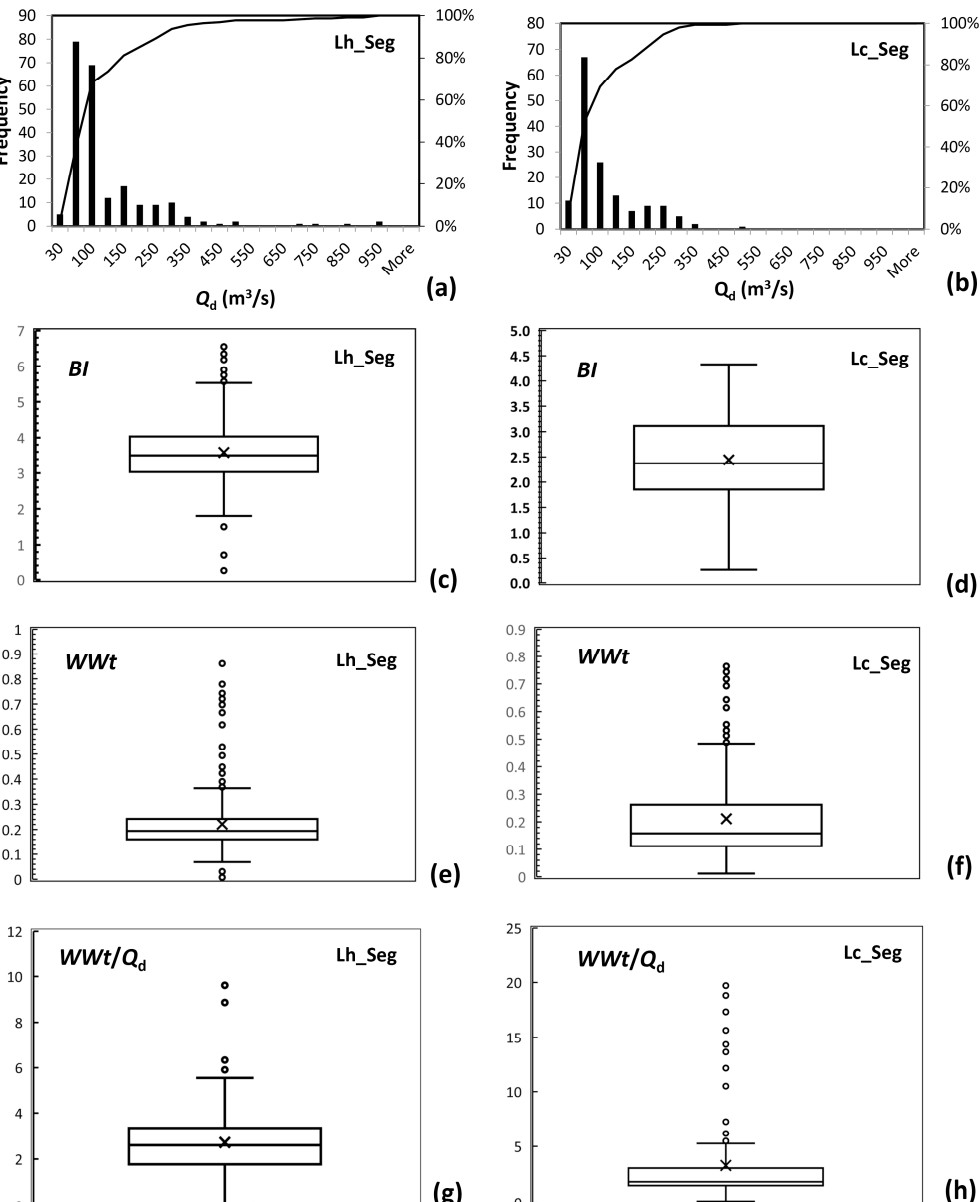

**Figure 6.** Statistical properties of the available daily discharges and obtained morphological indices in the two studied braided corridors. (**a**,**b**) Frequency and cumulative distribution of daily discharges ($Q_d$); (**c**,**d**) box plots of the braiding intensity ($BI$); (**e**,**f**) box plots of the total mean water width ($WWt$); (**g**,**h**) boxplots of the $WWt/Q_d$.

### 3.2.2. Temporal Trends of *WWt* Values and Their Relationship with Daily Discharges

Figure 7 shows the temporal evolution of the extracted *WWt* values in the two braided segments. The mean values of the *WWt* extracted from all three indices for the Lh_Seg were similar, which were 0.223, 0.221, and 0.245 km for NDWI, MNDWI, and NDMI, respectively (Figure 7a–c). Values of *WWt* for NDWI showed the least variation, marked by the highest value of 0.704 km (Figure 7a). The variation of *WWt* for MNDWI was higher. The maximum reached 0.861 km (Figure 7b). NDMI showed the highest variation of *WWt*, featuring the maximum value of 1.621 km (Figure 6c). This order of variations was further supported by the relative standard deviations (RSDs) of 0.391, 0.501, and 1.120 km for NDWI, MNDWI, and NDMI, respectively.

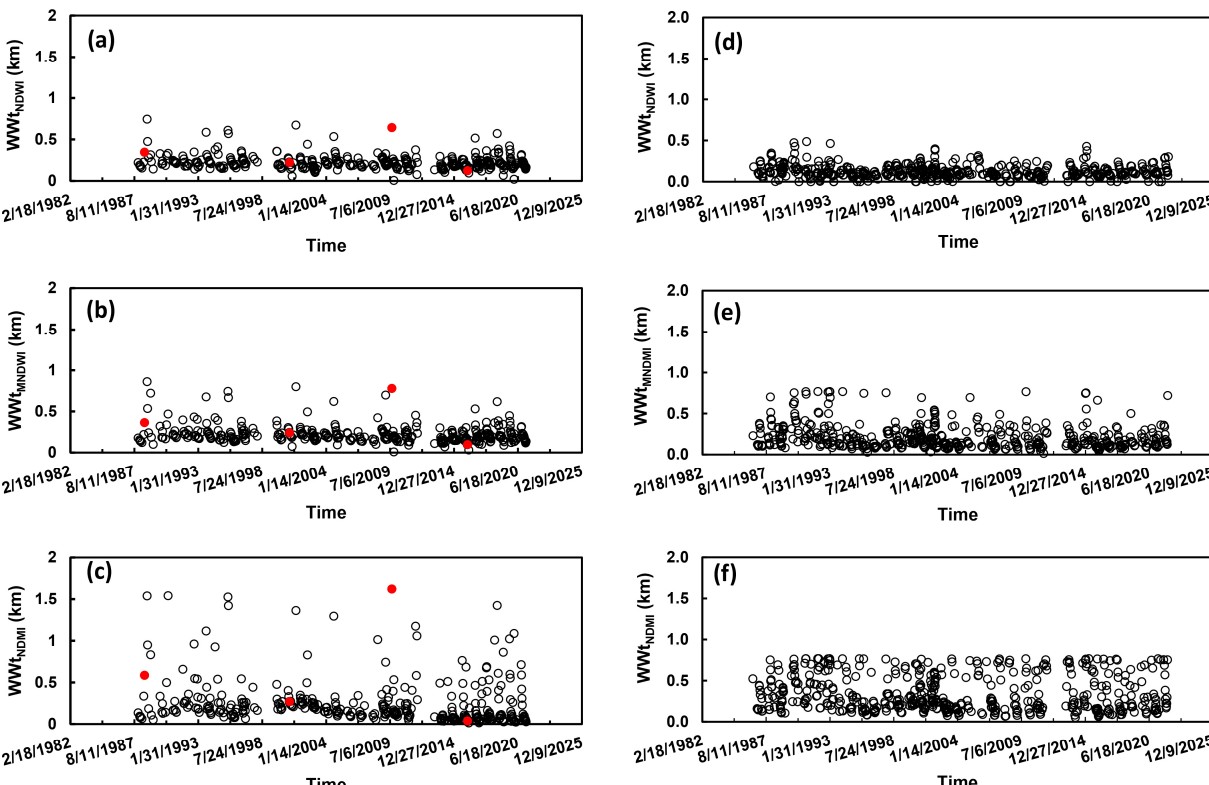

**Figure 7.** Temporal trends of the total mean water width (*WWt*) detected using the three selected water-detection indices for the two braided segments. (**a–c**) are for the Lh_Seg site, whereas (**d–f**) are for the Lc_Seg site. The four red points represent the four images that are associated with the four selected daily discharges.

For the Lc_Seg, the mean of *WWt* values for NDWI, which was 0.126 km, was much less than that of MNDWI and NDMI, which was 0.215 and 0.345 km, respectively. The variation of *WWt* values for NDWI, represented by RSD = 0.101 km, was the least among the three indices. MNDWI indeed had a higher variation of *WWt* (RSD = 0.119 km) than NDMI did (RSD = 0.105 km). The *WWt* values for both NDWI and MNDWI had higher degrees of variation in the Lc_Seg site than those in the Lh_Seg site.

The relationships between *WWt* and $Q_d$ with three indices for the two braided segments were analyzed using linear and power functions and for higher discharge values, respectively (Figure 8). In the Lh_Seg case, the linear function fits the data based on NDWI ($R^2 = 0.81$) better than the power one ($R^2 = 0.77$), mainly because of the better fittings for higher discharges, though both functions tend to overestimate the total channel widths for low discharges (Figure 8a). For MNDWI and NDMI, similar results prevailed, though NDMI seems to particularly overestimate widths for all discharges (Figure 8b,c). Among the three linear correlations, the one for *WWt* based on NDMI was affected by a higher scatter of the low and medium values. Except for some outliers in the range of smaller water discharges, the relationships between *WWt* and $Q_d$ for MNDWI and NDWI were similar. In the Lh_Seg site, the power of the fitted nonlinear line for all data based on MNDWI, which is 0.47, is less than that for higher discharges ($Q_d > 400$ m$^3$/s), which is 0.66 (Figure 8b). This suggests that the braided river in the Lh_Seg site is potentially highly supplied by sediment with relatively low channel banks, which facilitate the relatively faster increase in the total channel width as the discharge increases.

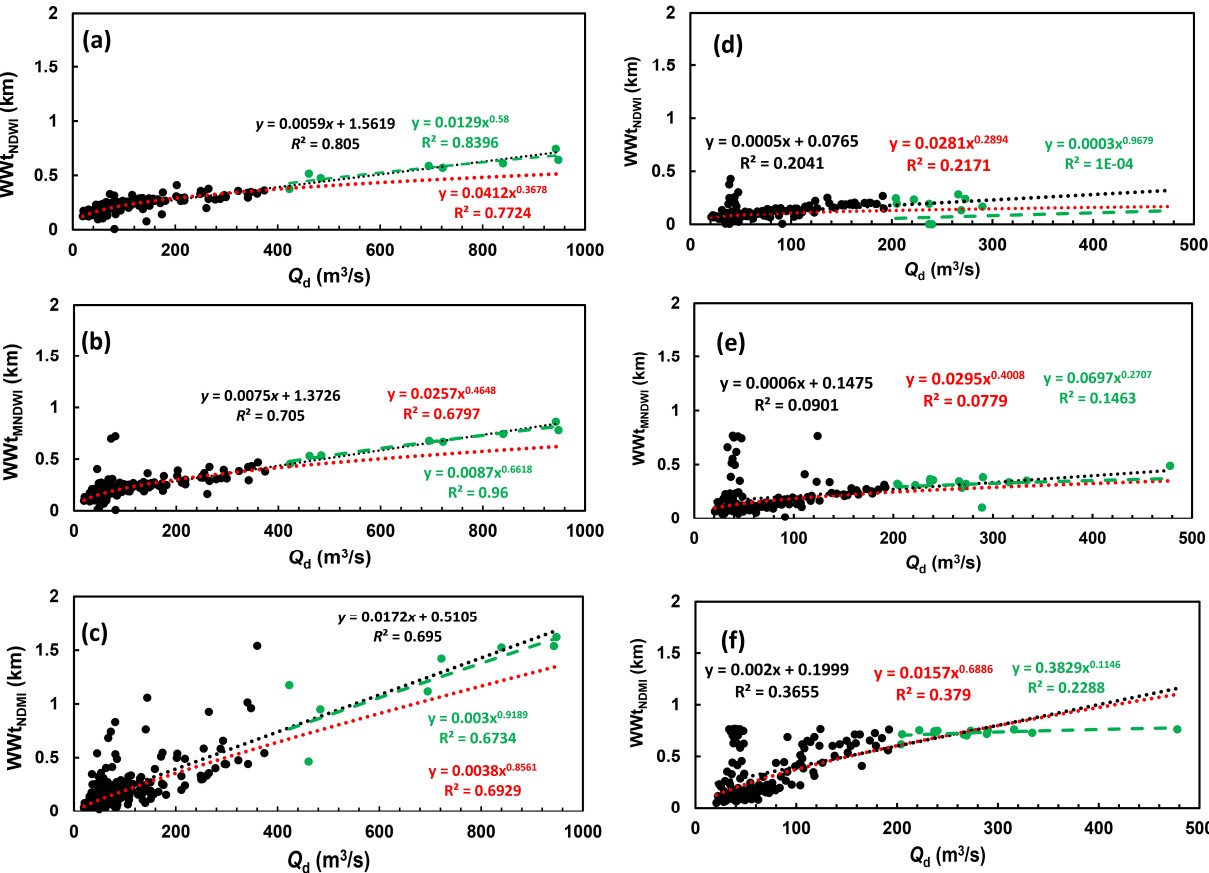

**Figure 8.** The relationship between the total mean water width (*WWt*) of the three selected water-detection indices and the daily discharge ($Q_d$) for the two braided segments. (**a–c**) are for the Lh_Seg site, whereas (**d–f**) are for the Lc_Seg site. The black trend line is a linear function, the red one is a power function for all data, while the green one is a power function for higher discharges only ($Q_d > 400$ m$^3$/s for the Lh_Seg site and $Q_d > 200$ m$^3$/s for the Lc_Seg site).

In the case of the Lc_Seg, the linear relationship fits all data as well as the power one, though both are not very strong for all three indices (Figure 8d–f), much lower than those in the Lh_Seg (Figure 8a–c). The very low $R^2$ values for NDWI and MNDWI were mainly caused by a series of high values in a very narrow range of discharges (Figure 8d,e). The correlations for NDWI and MNDWI are indeed stronger if these outliers are excluded ($R^2$ is 0.45 and 0.61, respectively). For NDMI, even after taking out these outlier points, the correlation would be still low because of the scatter of points in the range of medium and high discharges (Figure 8f). The nonlinear relationships in all three indices showed different patterns from those in the Lh_Seg site. The exponents of the power functions for all data are greater than those for higher discharge ($Q_d > 200$ m$^3$/s) for MNDWI and NDMI (Figure 8e,f). Although this exponent for NDWI is still higher for all data, the *WWt* values for higher discharges are not significantly higher than those with medium discharges (Figure 8d). These results suggest that the braided river in the Lc_Seg site tends to be less supplied by sediment with stronger elevation contrasts within the braided area.

Overall, values of *WWt* from both NDWI and MNDWI were closely related to water discharges in both braided rivers in the Lh_Seg and Lc_seg sites, whereas these correlations for *WWt* from NDMI in both sites were generally poor. The relatively weak relationships (both linear and power) with MNDWI than those with NDWI were mainly caused by the outliers in the range of low discharges (explained in discussion) (Figure 8). When they are removed, the R$^2$ of the linear relationship is 0.88 and 0.61 for the Lh_Seg and Lc_Seg, respectively, much higher than those of the linear relationships with NDWI (0.81 and 0.45). Therefore, MNDWI is best correlated with $Q_d$ once the outliers are removed. The different

patterns of the exponents in the power functions (Figure 8) imply that the braided river is apparently dominated by transport-limited processes in the Lh_Seg, while it tends to be limited by sediment supply in the Lc_Seg site. These different dominant sediment-transport processes may also explain why the river in the Lc_Seg site is less braided than that in the Lh_Seg site (see also Figure 6c,d).

The power of the fitted line for higher discharges is higher than that for all discharges (Figure 8d). For MNDWI and NDMI, the two types of power relationships are still relatively weak, though the power of the relationship for all discharges is higher than that for higher discharges only (Figure 8e,f), which is opposite to the case in the Lh_Seg site. Apparently, the braided river in the Lc_Seg site tends to be supply-limited. The different dominant sediment-transport processes may also explain why the river in the Lc_Seg site is less braided than that in the Lh_Seg site. The linear relationship fits all data as well as the power one, though both are not very strong for all three indices (Figure 8d–f).

### 3.2.3. Mean Water-Detection Indices at the Braided Corridor Scale and Their Relationships with Daily Discharges

In the Lh_Seg site, mNDWIcs remained roughly constant before 2012 with a mean of −0.0581 and RSD of 0.1620 (Figure 9a). After 2012, mNDWIcs dropped to −0.0871 with an RSD of 0.0846. This change was apparently consistent with the changed sensors of satellites in 2012, but the difference between the two mean values was not statistically significant. The values of mMNDWIcs showed a similar trend to those of mNDWIcs, featuring the mean of −0.0988 and −0.1207 for the two periods, respectively. Again, the difference between the two mean values was not statistically significant. However, they demonstrated much higher annual variations (RSD = 0.228 and 0.132 for the two periods, respectively) (Figure 9b). mNDMIcs displayed an opposite pattern, showing a lower mean of −0.0412 before 2012 and a higher one after 2012 −0.01313, respectively) (Figure 9c). However, mNDMIcs had higher degrees of annual variations in both periods (RSD = 0.534 and 0.489, respectively), compared with mNDWIcs.

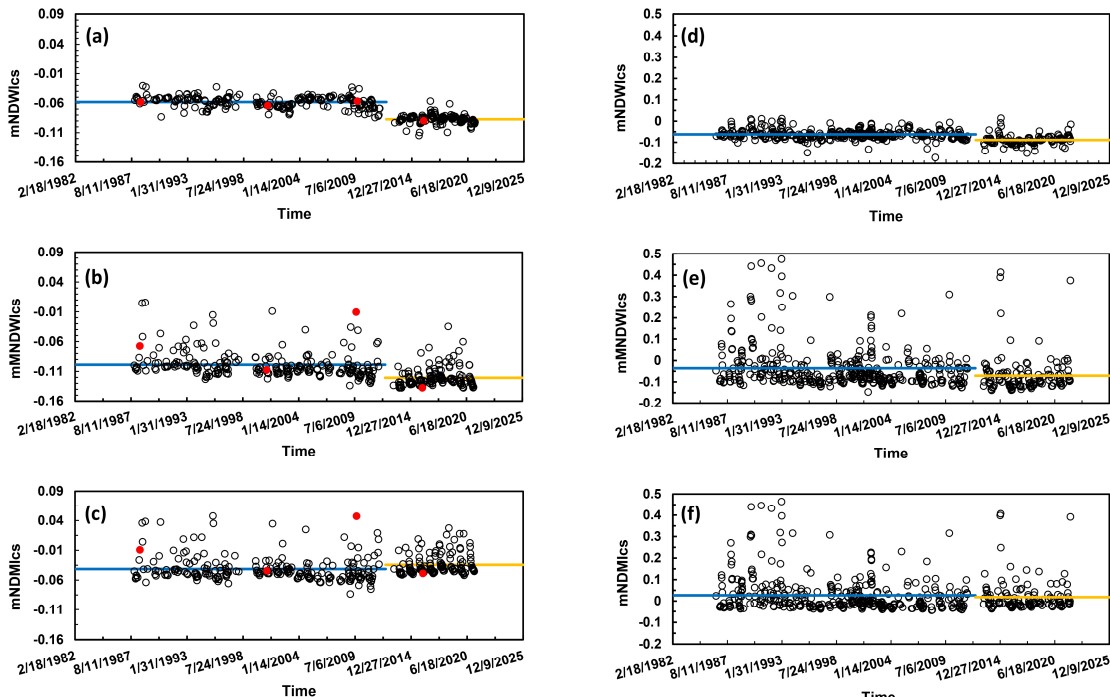

**Figure 9.** Temporal trends of the three mean indices calculated within each selected braided corridor. (**a**–**c**) are for the Lh_Seg site, whereas (**d**–**f**) are for the Lc_Seg site. The four red points represent the four images that are associated with the four selected daily discharges. The blue and yellow horizontal lines represent the mean of each index before and after 2012.

In the Lc_Seg site, the temporal change before and after 2012 was less discernable for mNDWIcs with means of −0.0624 and −0.0891 for the two periods, respectively (Figure 9d). Similar reduced differences of the means between the two periods were observed for mMNDWIcs (i.e., −0.0356 and −0.0709) and NmDMIcs (−0.0267 and −0.0182) (Figure 9e,f). Nonetheless, the annual variations of all three indices were discernably higher than those in the Lh_Seg. The RSD for mNDWIcs was 0.3482 and 0.2568, respectively. For mMNDWIcs, it became 2.7320 and 1.7226, respectively, whereas for mNDMIcs, it was even higher (i.e., 3.1185 and 4.0749, respectively). Apparently, the response of the three water-detection indices to surface water is different between these two sites.

When mean values of each index were plotted against the associated $Q_d$ in the Lh_Seg, mNDWIcs showed almost no correlation between the two (Figure 10a), but both mMNDWIcs and mNDMIcs were related to $Q_d$ (Figure 10b,c). This relation is not as strong as the *WWt* approach but is still significant. Between these two, mNDMIcs showed a better correlation with $Q_d$ ($R^2 = 0.563$) than mMNDWIcs did ($R^2 = 0.473$). In the Lc_Seg, mNDWIcs still did not have a clear correlation with $Q_d$ (Figure 10d), as in the Lh_Seg. Both mMNDWIcs and mNDMIcs were visually correlated with $Q_d$, but their $R^2$ values (i.e., 0.026 and 0.068) were very low (Figure 10e,f).

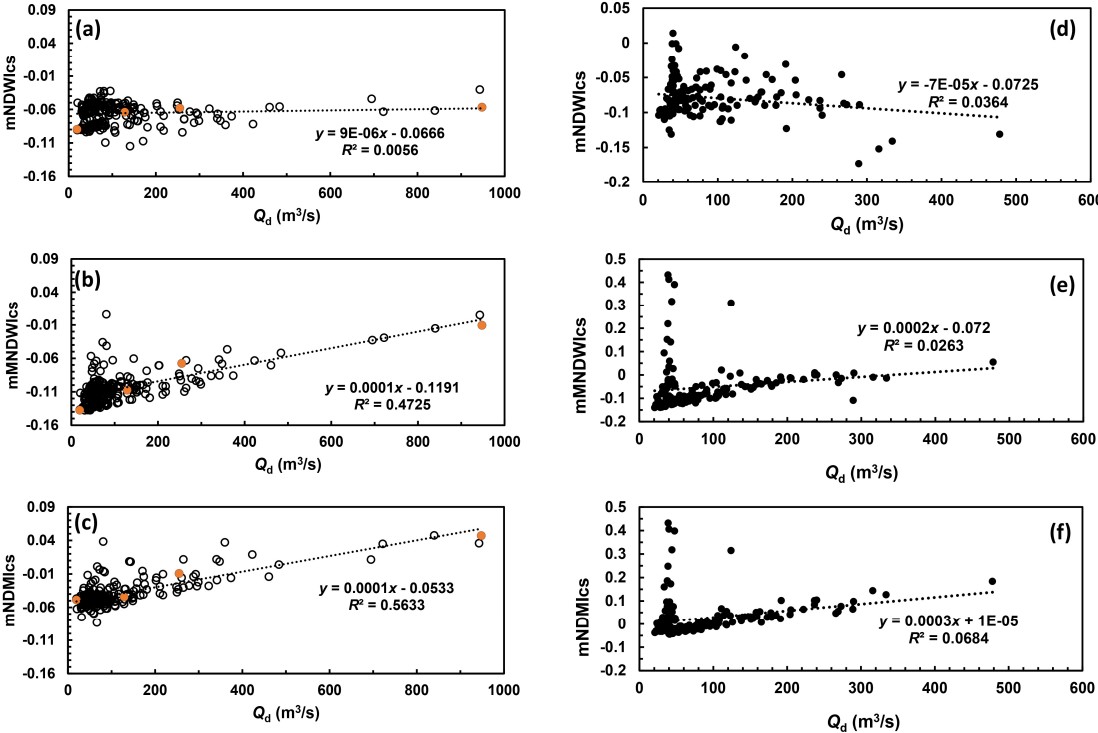

**Figure 10.** The relationship between the three mean indices calculated within each selected braided corridor and the daily discharge ($Q_d$) for the two braided segments. (**a**–**c**) are for the Lh_Seg site, whereas (**d**–**f**) are for the Lc_Seg site. The four orange points represent the four images that are associated with the four selected daily discharges.

## 4. Discussions

### 4.1. Pitfalls of Using WWt as a Proxy for the Water Discharges

The *WWt*–$Q_d$ relationship is potentially affected by the choice of the threshold to extract $A_s$, but we have shown that the threshold of 0 was the best option for extracting reasonable braided channel structure in the defined braided corridors (Figure 4). However, this would need to be verified in other braided segments.

The *WWt* extracted from MNDWI and NDWI showed similar strong relationships with $Q_d$ for both the Lh_Seg and Lc_Seg sites (Figure 8), whereas mean index values did not (Figure 10). What distinguishes the two is that *WWt* is obtained from $A_s$, which

reflects the number of pixels whose MNDWI and NDWI values are above the selected threshold for classifying water and non-water pixels, while mean index values are the actual reflectance values from the same set of pixels over the entire braided corridor. Thus, *WWt* obtained from $A_s$ hides the differences among index values within the same class (e.g., water) and is mainly affected by the threshold value selected subjectively. As $Q_d$ increases, the number of water pixels will always increase for a given threshold, regardless of the specific index values in the pixels. Therefore, *WWt* should be well related to the associated $Q_d$. However, *WWt* from NDMI is discernably less well correlated to $Q_d$, which is ascribed to the nature of NDMI being very sensitive to liquid water content and moisture conditions of vegetation [50,51]. This argument is further supported by the fact that the exponents of the power functions based on MNDWI for the two sites are well within the range of these exponents in reported hydraulic geometry in literature [37].

Additionally, *WWt* could be affected by the coarse resolution of images in terms of pixel sizes compared to the flow channel sizes. Small water channels below the pixel resolution (i.e., 30 m) cannot be classified as water (i.e., pixels with mixed water and not water components whose value is below the selected threshold for water class), and hence they are not counted in the $A_s$ calculation and then in *WWt*. On the other hand, an increased proportion of such small water channels with increasing discharge would cause an increase in the index value of all these mixed pixels and hence of the overall mean index value at the braided corridor scale. This resolution effect associated with $A_s$ detection, combined with the $A_s$ dependence on the index threshold, introduces uncertainties to the *WWt*–$Q_d$ relationship, notably for low discharges. Figure 8 clearly shows a more significant variability for discharge below 200 m$^3$/s and even greater below 100 m$^3$/s in the two braided segments. Using MNDWI-based results as an example, we calculated the residual of *WWt* between the measured and predicted from the *WWt*–$Q_d$ relationship, after eliminating the outliers for the lower discharges in the two braided corridors (Figure 11a,b). We then plotted the residuals against the associated *BI* values (Figure 11c,d), which clearly show that *WWt* residuals are strongly related to *BI* for lower discharges, particularly for the Lh_Seg site. This supports the existence of the size effect, linked to image resolution, which explains the variability of measuring $Q_d$ from $A_s$ derived width, specifically in the case of braided river corridors with an intense braiding pattern (i.e., many and little-sized channels such as the Lh_Seg site) (see also Figure 6).

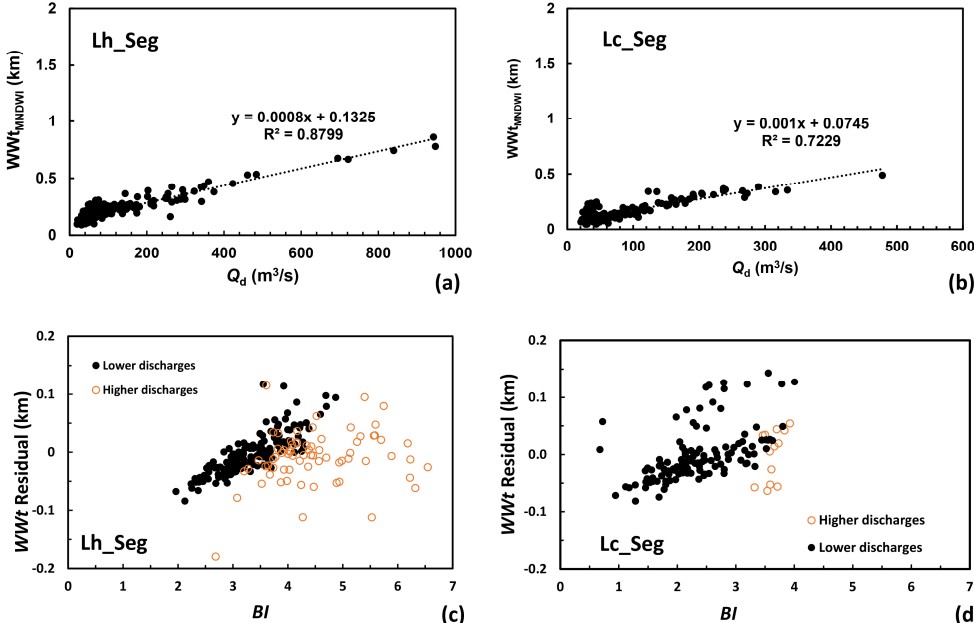

**Figure 11.** The impact of the spatial resolution of the images on extracted channel morphological indices. (**a**,**b**) The true (and stronger) correlation between *WWt* calculated using MNDWI and the associated daily

discharge ($Q_d$) in the two braided corridors after eliminating the outliers. (**c**,**d**) show the relationship between *BI* and the *WWt* residuals calculated as the difference between measured *WWt* values and the predicted ones using the relationships shown in (**a**,**b**). Lower discharges refer to $Q_d \leq 100 \text{ m}^3/\text{s}$, while high discharges denote $Q_d > 100 \text{ m}^3/\text{s}$.

### 4.2. Mechanisms of Index Performance in Relating to Water Discharges

In both selected sites (i.e., Lh_Seg and Lc_Seg), a set of off-trend points in the low but narrow range of discharges had abnormally high index values for *WWt* obtained from $A_s$ (Figure 8b,c,e,f) and mMNDWIcs and mNDMIcs (Figure 10b,c,e,f). These points are associated with the discharges that occurred in winter when temperature was often below the frozen threshold during which the braided channels were partly or mostly covered by ice or snow. Taking out these 'abnormal' points, the correlation strength increases significantly. In Lh_Seg, the $R^2$ value becomes 0.880 for mMNDWIcs and 0.703 for mNDMIcs, respectively. In Lc_Seg, the $R^2$ value is improved from 0.090 to 0.723 for mMNDWIcs and from 0.379 to 0.631 for mNDMIcs, respectively. In both sites, the correlation is generally stronger for mMNDWIcs.

The physical characteristics of reflectance ($\rho$) for individual bands G, NIR, and Swir (see Figure 3 in Huang et al., 2018 [10]) suggest that $\rho_G$—$\rho_{NIR}$ is less than $\rho_G$—$\rho_{Swir}$ for the same image covered by snow/ice, and $\rho_G$ is much higher than $\rho_{NIR}$ or $\rho_{Swir}$ for snow. Therefore, the same image covered by snow/ice would lead to a higher value of mMNDWIcs than that of mNDWIcs and mNDMIcs. This effect was demonstrated for the Lc_Seg in our earlier work [4]. Two sets of images displaying this effect in the Lh_Seg are also provided in this study (Figure 12).

Our analyses suggest that the generally poor correlation between mNDWIcs and the associated $Q_d$ is mainly caused by (1) the lower sensitivity of mNDWIcs to water, particularly at higher discharges, which might be related to higher turbidities in these flows, and (2) the stronger responses of NIR to increased vegetation than those of green to increased water surface area due to increased $Q_d$ values.

Our results signified that mMNDWIcs is the best water-detection index that can be used to establish the rating curve with $Q_d$ at the reach scale and within the braided corridor. Surprisingly, even if it is still difficult to interpret and understand, mNDMIcs seems also a good $Q_d$ predictor for the two reaches. Moreover, Figure 9 also shows that mNDMIcs was much less affected by the sensor changes than mMNDWIcs in 2012.

The increase in $Q_d$ is always associated with the increase in temperature in both study sites (i.e., the wet season), which encourages vegetation development, leading to a higher density and coverage of vegetation. As such, some vegetated pixels with higher moisture contents may also be classified into the water group based on the selected threshold, which might weaken the relationship between WWt from NDMI and $Q_d$. Because the increase in the mNDMIcs value reflected not only the increased value of $Q_d$ but also the increased water content in the also increased vegetation coverage, the strong correlation between mNDMIcs and $Q_d$ (Figure 10c,f) did not specifically characterize the relationship between $Q_d$ and the surface water caused by $Q_d$. Further tests are needed to analyze the behavior of the NDMI index in braided systems with abound riparian and bar vegetation in other regions.

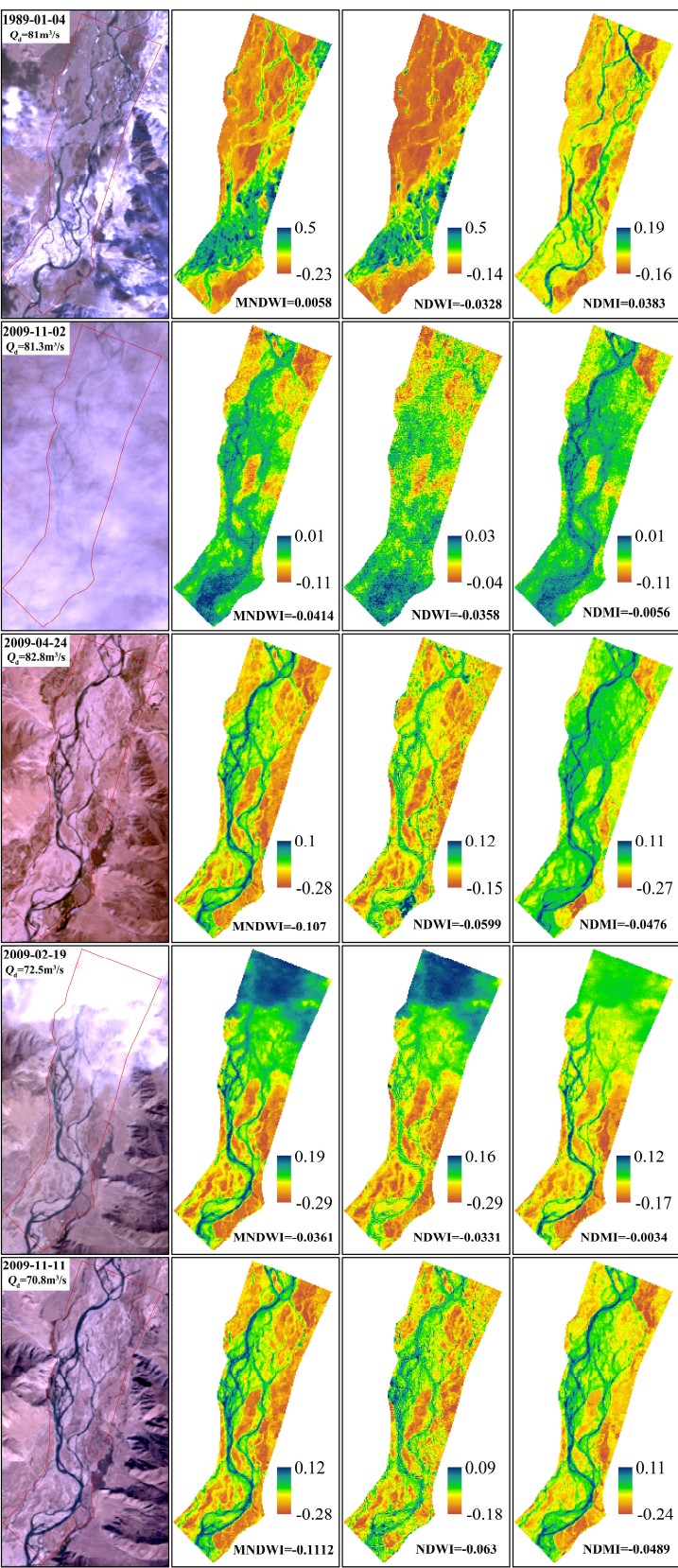

**Figure 12.** Examples of images with similar water discharges but very different index values. Images in the first, second, and fourth rows were covered partly by snow or ice, whereas those in the third and fifth rows were under normal conditions. The values of the three selected indices in the latter are significantly higher than those in the former.

## 5. Conclusions

The availability of cloud-based computer platforms has made it possible to use large numbers of remotely sensed images to reveal historical changes in the braided river morphology. Based on a time series of three water-detection metrics (i.e., NDWI, MNDWI, and NDMI) and the associated total water width obtained from surface-water areas extracted from the two braided reaches using Google Earth Engine, we examined the characteristics of the three indices and their performance of linking to the associated water discharges. Although *WWt* may be well correlated to water discharges, these were determined in terms of the pre-set thresholds of the indices, which are subject to errors due to uncertainties in the selection of the threshold for a given index. Directly linking the mean index values calculated in braided corridors to water discharges has the advantage of avoiding these uncertainties.

mNDMIcs apparently is not well correlated to water discharges, whereas mNDWIcs and mMNDWIcs performed well when used to establish the discharge rating curve in terms of either their values or the extracted surface-water areas from them. The performance of mNDMIcs still needs to be interpreted and tested in other contexts. The fundamental mechanism driving the poor performance of mNDWIcs ties to the physical characteristics of the near-infrared band whose strong responses to increased vegetation coverage offsets the positive response of the green band to the increased water discharges, leading to unchanged or even decreased mNDWIcs values as $Q_d$ increases.

The remote locations and harsh physical conditions in the Tibet plateau make it difficult to deploy frequent field measurements for obtaining continuous hydrological changes. As such, only limited state-run gauging stations have been established with limited discharge data available. Because the relationship between mMNDWIcs and $Q_d$ is linear, it is easy to convert it to a predictive equation, such that mMNDWIcs values in the periods of no measured $Q_d$ values can be used to allow for a rough but almost continuous estimate of discharge conditions and their variation within the different braided reaches of the plateau, assuming no braided planform changes occurred.

Although the mMNDWIcs–$Q_d$ relationship is robust in the two selected braided reaches, it may be affected by physiographic conditions in different regions, which likely produce some 'outliers' that need to be removed, just like the classic sediment rating curves [52]. However, being able to generate an mMNDWIcs–$Q_d$ relationship is potentially valuable for studying morphodynamic processes in mid-sized braided rivers, particularly those located remotely with limited hydrological data (e.g., in Balkin, India, and Andres Plateau). In such a domain, the comparison of *WWt* with the *BI* among the braided reaches and through time may be also interesting to detect different braided types in the plateau and potentially some changes through time if some are affected by some changes in their drivers (water and sediment supply).

**Author Contributions:** Conceptualization, P.G., B.B. and H.P.; Methodology, P.G., B.B. and H.P.; Software, B.B.; Validation, P.G., B.B. and H.P.; Formal analysis, P.G.; Investigation, B.B. and H.P.; Data curation, P.G. and Y.Y.; Writing—original draft, P.G.; Writing—review & editing, P.G., B.B. and H.P.; Visualization, Y.Y.; Project administration, H.P.; Funding acquisition, P.G., H.P. and Z.L. All authors have read and agreed to the published version of the manuscript.

**Funding:** This work has been supported by the EUR H$_2$O'Lyon (ANR-17-EURE-0018) of Université de Lyon (UdL), within the program 'Investissements d'Avenir' operated by the French National Research Agency (ANR), National Natural Science Foundation of China (U2243214), Major Water Resources Science and Technology Project of Hunan Province (XSKJ2022068-12), and Key R&D Program of Hubei Province (2023BCB110).

**Data Availability Statement:** The data presented in this study are available on request from the corresponding author.

**Conflicts of Interest:** The authors declare no conflict of interest.

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
