# Peer review of "Can Water-Detection Indices Be Reliable Proxies for Water Discharges in Mid-Sized Braided Rivers Using Coarse-Resolution Landsat Archives?"

_remotesensing, doi:10.3390/rs16010137_

Round 1

Reviewer 1 Report

Comments and Suggestions for Authors

In the presented work, the authors compare the physical characteristics of three most commonly used water-detection (WD) indices, NDWI, MNDWI and NDMI, for two mid-sized braided reach segments in the Qinghai-Tibet Plateau in China that have different morphological structures. Based on the Google Earth Engine web-interface, the authors calculate the total mean water width (WWt) and braiding index (BI), as well as the mean values of these indices over four decades at the braided corridor scale.

The manuscript presents new and interesting data and findings and is altogether well written and well presented. The addressed topic is of relevance and interest and fits into the scope of the journal Remote Sensing.

I have only a few comments/questions for possible minor revisions:

- Page 3, lines 137-139: Please provide more details and be more precise here.

- Page 5, lines 172ff: Please provide more details on potential errors of the calculated WWt and BI.

- Page 19, lines 542-544: Please provide more details and be more precise on the possible effects here.

- Please highlight more clearly which wider implications your interesting findings have. 

Author Response

- Page 3, lines 137-139: Please provide more details and be more precise here.

Response: we added some detailed information here (see lines 137-139 in the annotated version of the paper)

- Page 5, lines 172ff: Please provide more details on potential errors of the calculated WWt and BI.

Response: The reviewer is right to highlight the errors issue. For such metrics the errors are directly related to the water body extraction method, which is based on the threshold method and on index value adopted (here is water index > 0), as well as on the resolution of the source images (and the associated detection issues and mixed pixel issues). These points are discussed later in the MS. However we can state that errors for both indices can be considered negligeable since these are spread out at the reach scale and can be considered insignificant for the purposes of the work. Indeed, the aim is not to specifically detect the water surface but to explore the relationship with Q. As well, the observed relationship and associated coefficient are consistent with previous works about the hydraulic geometry of braided rivers. We’ve added a sentence at the end of the paragraph to address the error issue (see lines 193-196 in the annotated version of the paper).

- Page 19, lines 542-544: Please provide more details and be more precise on the possible effects here.

- Please highlight more clearly which wider implications your interesting findings have. 

Response: we provided additional information in this section (see Line 597-606).

Reviewer 2 Report

Comments and Suggestions for Authors

 This paper submitted to the Remote Sensing journal explores various indices for water and non-water identification, determining optimal threshold values for channel structure. NDWI and MNDWI effectively detect braided channels with a threshold of zero. Additionally, WWt correlates better with Qd linearly, especially when derived from MNDWI. At the braided corridor scale, mMNDWIcs shows a stronger relationship with Qd than mNDMIcs.

While well-structured and clear, the paper requires moderate revisions before publication:

1)     Clarify if data normalization, crucial for cross-sensor comparability, was performed using Landsat 4, 5, 7, and 8. Cross-validate using reference data to ensure effective normalization, especially in Google Earth Engine (GEE).

2)     Specify whether analyses were conducted under bankfull or low waters conditions. Provide details on why specific flow variations (19.6, 129, 255, and 948 m³/s) were selected based on river hydrograph examination. How was the selection ?

3)     Correct "Swir" to "SWIR" in Line 173.

4)     Enhance the quality of figures (1 and 2) with more than 300ppi.

5)     Address how the script in GEE handles spectral mixing challenges in a 30x30m area with various endmembers (water, vegetation, soil). Clarify the approach to managing spectral mixture in GEE.

Author Response

1)     Clarify if data normalization, crucial for cross-sensor comparability, was performed using Landsat 4, 5, 7, and 8. Cross-validate using reference data to ensure effective normalization, especially in Google Earth Engine (GEE).

Response: As shown in Figure 9, the variability of the spectral indices is constant through time, even for major sensors changes before and after 2012 (changes are not significant). It is worth mentioning that we have also carried out visual check and expert-based interpretation of the quality of time series images and of the extracted indices according to visual analytics technics (see for instance: https://kops.uni-konstanz.de/server/api/core/bitstreams/f09ef58c-0acc-47c2-a045-f16d852a2229/content).

2)     Specify whether analyses were conducted under bankfull or low waters conditions. Provide details on why specific flow variations (19.6, 129, 255, and 948 m³/s) were selected based on river hydrograph examination. How was the selection ?

Responses: The aim of this selection was to represent a wide range of discharge conditions. We explained the nature of these discharges and the reason of their selection (see lines 206-208 in the annotated version of the paper).

3)     Correct "Swir" to "SWIR" in Line 173.

Response: We did it (see lines 164-165, 173 in the annotated version of the paper).

4)     Enhance the quality of figures (1 and 2) with more than 300ppi.

Response: the current Figure 1 has already got a resolution of 800 dpi.

 5)     Address how the script in GEE handles spectral mixing challenges in a 30x30m area with various endmembers (water, vegetation, soil). Clarify the approach to managing spectral mixture in GEE.

Response: We are aware of the spectral mixing issues, notably while working with images whose pixel size is 30x30 m. This is why (i) we tested different threshold values to use the most appropriate to extract water bodies in the study sites (see Figure 4); and (ii) we claim for the use of mean values of spectral indices at river corridor scale, which is independent from the thresholding issues (see Figure 10). These rely on the hypothesis that higher values of the indices are linked to the presence of water, according to the existing literature. However, it is worth mentioning that the relationship between WWt (notably obtained from the extraction of water bodies from MNDWI with a threshold of 0) and Q provides results consistent with previous work on hydraulic geometry of braided rivers (see lines 493-496 in the annotated version of the paper).

Reviewer 3 Report

Comments and Suggestions for Authors

Reviewer report: “Can water-detection indices be reliable proxies for water discharges in mid-sized braided rivers using coarse resolution Landsat archives?

By Peng Gao, Barbara Belletti, Herve Piegay, Yuchi You, Zhiwei Li

The generation of relationships between braided corridor water-detection indices and water discharges using Landsat archives provides a method for effectively and quickly obtaining continuous water discharges for a better understanding of the morphodynamics of braided rivers.

The goals of this research from this manuscript are as follows:

1) analyzing the relationship between mean water width and different mean water-detection indices computed within braided corridors using conventional water surface area thresholding across a variety of water discharges;

2) determining water-detection index is most closely associated with water discharges;

3) interpreting potential differences between these indices.

Important new information about braided river systems can be gained from this research in terms of both scientific knowledge and real-world applicability. Investigating morphodynamic processes in medium-sized, braided rivers that have minimal hydrological data should be easier by creating a mMNDWIcs-Qd relationship (the mean values of Modified Normalized Difference Water Index over about four decades at the braided corridor scale-daily discharge). Comparing the total mean water width with the braiding index among the braided reaches and over time in this domain may be useful to discern between various braided kinds on the plateau and maybe discover changes over time if some are impacted by changes in their drivers (water and sediment supply). Better monitoring and management of these dynamic environments is the overarching objective, and it is in line with that purpose.

Minor observation: Hydraulic geometry (Leopold and Maddock, 1953, https://doi.org/10.3133/pp252) refers to the power laws relating the channel width W, mean depth D, and mean velocity V to discharge Q: W = aQb, D = cQf, and V = kQm. Since this method is explained in the introduction of the paper, all the graphs in Figures 8 and 11 must display correlations of the power equation kind in addition to linear ones because hydraulic geometry is unfamiliar with linear correlations.

The manuscript is clearly presented, well written and illustrated. The bibliography used is sufficient. The topic is interesting, and the author has done a great job of realizing the subject. The paper should be published because it contains material that the journal's readers will find of interest. This analysis is a complex and laborious one, and the conclusions are pertinent.

Best wishes,

The reviewer

Author Response

Minor observation: Hydraulic geometry (Leopold and Maddock, 1953, https://doi.org/10.3133/pp252) refers to the power laws relating the channel width W, mean depth D, and mean velocity V to discharge Q: W = aQb, D = cQf, and V = kQm. Since this method is explained in the introduction of the paper, all the graphs in Figures 8 and 11 must display correlations of the power equation kind in addition to linear ones because hydraulic geometry is unfamiliar with linear correlations.

Response: This is a very good point. We took the reviewer’s point and fitted power functions to both higher discharges and all discharges. We then compared the results between the two types of the fitted lines (linear vs nonlinear) and showed that the linear relationship performed better. Therefore, the linear relationship is used for calculation in the remaining part of the paper This means that we do not have to change Figure 11 as suggested by the review. But this very useful comment induces a few adjustments and clarifications both in the result and discussion parts (see lines 364-395 in the annotated version of the paper).
